# 1*H*-Imidazole-2,5-Dicarboxamides as NS4A Peptidomimetics: Identification of a New Approach to Inhibit HCV-NS3 Protease

**DOI:** 10.3390/biom10030479

**Published:** 2020-03-21

**Authors:** Abdelsattar M. Omar, Mahmoud A. Elfaky, Stefan T. Arold, Sameh H. Soror, Maan T. Khayat, Hani Z. Asfour, Faida H. Bamane, Moustafa E. El-Araby

**Affiliations:** 1Department of Pharmaceutical Chemistry, Faculty of Pharmacy, King Abdulaziz University, Alsulaymanyah, Jeddah 21589, Saudi Arabia; asmansour@kau.edu.sa (A.M.O.); mkhayat@kau.edu.sa (M.T.K.); 2Pharmaceutical Chemistry Department, Faculty of Pharmacy, Al-Azhar University, Nasr City, Cairo 11884, Egypt; 3Department of Natural Products and Alternative Medicine, Faculty of Pharmacy, King Abdulaziz University, Alsulaymanyah, Jeddah 21589, Saudi Arabia; melfaky@kau.edu.sa; 4Computational Bioscience Research Center, Division of Biological and Environmental Sciences and Engineering, King Abdullah University of Science and Technology, Thuwal 23955, Saudi Arabia; stefan.arold@kaust.edu.sa; 5Center for Scientific Excellence Helwan Structural Biology Research (HSBR), Faculty of Pharmacy, Helwan University, Ain Helwan, Cairo 11795, Egypt; sameh_soror@pharm.helwan.edu.eg; 6Department of Biochemistry and Molecular Biology, Faculty of Pharmacy, Helwan University, Ain Helwan, Cairo 11795, Egypt; 7Department of Medical Microbiology and Parasitology, Faculty of Medicine, King Abdulaziz University, Jeddah 21589, Saudi Arabia; hasfour@kau.edu.sa; 8Department of Biochemistry, Faculty of Medicine, King Abdulaziz University, Jeddah 21589, Saudi Arabia; fbamanea@kau.edu.sa

**Keywords:** NS3 inhibitors, allosteric inhibitors, NS4A, peptidomimetics, imidazole, hepatitis C virus, molecular dynamics, *Flaviviridae*, DSLS, binding assay

## Abstract

The nonstructural (NS) protein NS3/4A protease is a critical factor for hepatitis C virus (HCV) maturation that requires activation by NS4A. Synthetic peptide mutants of NS4A were found to inhibit NS3 function. The bridging from peptide inhibitors to heterocyclic peptidomimetics of NS4A has not been considered in the literature and, therefore, we decided to explore this strategy for developing a new class of NS3 inhibitors. In this report, a structure-based design approach was used to convert the bound form of NS4A into 1*H*-imidazole-2,5-dicarboxamide derivatives as first generation peptidomimetics. This scaffold mimics the buried amino acid sequence Ile-25` to Arg-28` at the core of NS4A_21`–33`_ needed to activate the NS3 protease. Some of the synthesized compounds (Coded MOC) were able to compete with and displace NS4A_21`–33`_ for binding to NS3. For instance, *N*^5^-(4-guanidinobutyl)-*N*^2^-(*n*-hexyl)-1*H*-imidazole-2,5-dicarboxamide (MOC-24) inhibited the binding of NS4A_21`–33`_ with a competition half maximal inhibitory concentration (IC_50_) of 1.9 ± 0.12 µM in a fluorescence anisotropy assay and stabilized the denaturation of NS3 by increasing the aggregation temperature (40% compared to NS4A_21`–33`_). MOC-24 also inhibited NS3 protease activity in a fluorometric assay. Molecular dynamics simulations were conducted to rationalize the differences in structure–activity relationship (SAR) between the active MOC-24 and the inactive MOC-26. Our data show that MOC compounds are possibly the first examples of NS4A peptidomimetics that have demonstrated promising activities against NS3 proteins.

## 1. Introduction

Hepatitis C is a life-threatening, widespread viral infection [1]. The virus initially infects liver cells though remains asymptomatic for extended periods. This invisibility constitutes epidemiological challenges as patients diagnosed with hepatitis C come from two pools: recently infected and late-stage asymptomatic. The initial symptoms, such as fever, fatigue, nausea, and liver tenderness, can be misleading, as they are tolerated by most patients. After several years of this blood-borne infection, the virus activates, replicates, and causes complications that start with liver scarring, fibrosis followed by cirrhosis and, eventually, liver failure and carcinoma [2,3]. Awareness of hepatitis C health problems and forceful efforts to combat the spread of hepatitis C virus (HCV) in healthcare settings have led to a significant decline in new hepatitis C cases. However, the annual death toll from complications of this disease remains as high as 400,000 due to the above described unique clinical vs. epidemic profile of the chronic hepatitis C. An estimated 70 million people globally are currently infected with HCV, constituting a major health problem [1]. In 2013, the death rate from HCV complications surpassed that of HIV. In 2015, hepatitis C-related deaths exceeded that of tuberculosis and malaria combined [4].

Up until 2013, interferon (or pegylated interferon) and ribavirin were the most effective available therapies, but their long-term efficacies were as low as 3% [5]. Identification of the hepatitis C genome–proteome in the early 1990s prompted intensive research that led to the introduction of the new class of direct antiviral agents (DAAs). DAAs revolutionized the treatment of hepatitis C and succeeded to control the global crisis of this epidemic that was once described as the “Third Millennium Challenge” [2]. Recently, combinations of DAAs have become an effective way to cover hepatitis C infection of different genotypes [6,7]. Nonetheless, according to the World Health Organization, there are several remaining challenges that must be met to eradicate HCV [1]: (1) access to the treatment in economically challenged areas, which must be increased [8]; (2) emerging drug resistance, which is an expected problem even with combination therapy [9,10]; and (3) efficacy in broader sectors of HCV patients, such as the elderly and impaired kidney and liver patients, which must be improved [11]. Thus, there is a clear need to identify new approaches to inhibit viral targets and to increase the DAA arsenal [12].

The HCV genome is a positive-sense, single-stranded RNA virus belonging to *Flaviviridae* [13]. The translated virus genome mainly corresponds to structural and nonstructural proteins. The nonstructural (NS) proteins include NS3, NS4A, NS4B, NS5A, and NS5B, which comprise factors essential for the maturation and replication of the virus [14]. NS3 dually functions as a protease (N-terminal domain) and an RNA helicase (C-terminal domain) [15]. When activated as a protease, it catalyzes the processing of the viral proteome to functional proteins by cleaving NS3–NS4A, NS4A–NS4B, NS4B–NS5A, and NS5A–NS5B junctions. The proper folding of NS3 is indispensable for protease activity and occurs only when it binds to NS4A, a 54 amino acid peptide with multiple functions. NS4A is not only required for NS3 activation [16,17] but is also important for the integration of NS3 into the host cell endoplasmic reticulum [18] and for neutralizing the host cell immune responses [19,20]. The NS3/4A protease was the first and foremost targeted viral protein with DAAs that bind to its substrate site [21]. Interference with the NS4A binding site, on the other hand, has not been evaluated thoroughly as a mechanism to allosterically inhibit the NS3 protein, especially by drug-like peptidomimetics. In this article, we present compounds that were designed to compete with and replace NS4A on its NS3 binding site, leading to NS3 inhibition [22]. 

## 2. Results

### 2.1. Rationale and Design

The NS4A N-terminal intercalates between the β-strands A0 and A1 (the first 28 residues of the N-terminal) of NS3. The association with NS4A induces proper conformation of the apoenzyme and increases the proteolytic activity of NS3 by ~950 times [14,16,23]. Accumulated evidence established that the central region of NS4A (Gly21`-Leu32`) is sufficient for activation of the NS3 protease [24,25,26,27]. Throughout this paper, the prime (`) mark is applied to differentiate the residues of NS4A from that of the NS3 apoenzyme. It has also been confirmed by our laboratory [28,29] and others [26,30] that certain mutants can bind to the NS4A site and inhibit the protease function of NS3. In their reporting of the first crystal structure of NS3/4A, Kim and coworkers stated that “the contact surface between NS3 and NS4A is quite extensive and provides another possible site for the design of anti-HCV chemotherapeutic agents” [14]. However, subsequent research focused merely on the discovery of NS3/4A substrate site inhibitors, while mention of the NS4A site inhibitors is rare, with only a few reports describing a hypothetical approach [14,31,32]. Accordingly, we decided to pursue this concept by designing peptidomimetics that bind and replace NS4A on the protease domain of NS3.

The first step in de novo design of NS4A peptidomimetics was to inspect the crystal structures of NS3–NS4A complexes (Protein Data Bank (PDB) Code: 1NS3) [23]. We noticed that NS4A forms a β-stand that is mostly extended, except in a turn (kink) featuring a nearly planar area. This planar kink is composed of one eclipsed *cis* bond with a dihedral angle of −12° (the negative sign denotes a deviation to the opposite side of the Val-26` side chain) and four near perfect planar *trans* bonds spanning through Val-26` to Arg-28` (Figure 1A, torsion table in Appendix A). This *cis* bond largely depends on the presence of Gly-27` as it allows lessening the steric conflict with the Val-26`side chain. 

Interestingly, this glycine turn is conserved in all crystal structures of HCV NS3 protease in the Protein Data Bank (PDB) deposited to date, regardless of their genotype [33,34,35] or of the sequence constructs used (NS4A fused with NS3 or non-fused) [14,23,35] (see Appendix A). Accordingly, we postulated that an imidazole nucleus appending two amide groups should mimic this planar region. Docking experiments by manual visually guided methods suggest that the 1*H*-imidazole-2,5-dicarboxamide nucleus can occupy the planar region while keeping key interactions of NS4A in place without posing significant steric conflicts (Figure 1B). In particular, the N1 of the imidazole acted as a hydrogen bond donor with the backbone carbonyl of Ile-35 (located in the A1 β-sheet), which also made a hydrogen bond with CO of the 5-carboxamide group of the designed scaffold. The carbonyl of the C2-carboxamide group occupied a planar area in which Val-26` rests. 

In previous studies, the Val-26` side chain was deemed unnecessary for NS4A function and, therefore, could be safely replaced by the intended amidic carbonyl [26]. The pocket accommodating the Ile-25` hydrophobic side chain has been sought as an opportunity to increase the binding potency of our compounds. In the context of probing this hydrophobic pocket, we decided to synthesize compounds of different hydrophobic substituents at the C2 amide group. The imidazole’s C5 carboxamide occupied the same position of the amide bond between Gly-27` and Arg-28`, conserving a hydrogen bond with Gln-8 (located within the A0 sheet). Its agmantyl substituents occupied the same position of the Arg-28` side chain. In addition, the bifurcate hydrogen bonding between the backbone carbonyl of Val-26` and NH groups of Thr-10 and Arg-11 were compensated by hydrogen bonding accepted by the N3 of the 1*H*-imidazole core (Figure 2). 

### 2.2. Synthesis of 1H-Imidazole-2,5-Dicarboxamide Derivatives

The final MOC compounds were synthesized using reported chemistry [36,37,38,39] as illustrated in synthetic Scheme 1. MOC refers to the project code and, therefore, final compound coding is not necessarily sequential not starting from MOC-1. Starting from cyanoformic acid ethyl ester (**1**), the nitrile group was converted to the *N*-hydroxyamidino derivative **2** by nucleophilic addition of hydroxylamine [38]. The pivotal imidazole intermediate **3** was prepared by thermal microwave irradiation (MWI)-induced cyclocondensation of **2** with *tert*-butyl propiolate [36,37]. This compound features the desired imidazole nucleus containing two different ester groups: ethyl (alkali labile) and *t*-butyl (acid labile). Therefore, trifluoroacetic acid (TFA) treatment could selectively hydrolyze the *tert*-butyl ester to imidazole-5-carboxylic acid **4**. 

The carboxylic acid was reacted with selected amines of R^2^ substituents under catalysis of 1-ethyl-3-(3-dimethylaminopropyl)carbodiimide (EDCI)/1-hydroxybenzotriazole hydrate (HOBt) to give the intermediates **5a**–**c**. The ethyl esters **5a**–**c** were hydrolyzed into the lithium salts **6a**–**c** using LiOH. The final compounds were obtained from **6a**–**c** by coupling with the desired amines (R^1^-NH_2_) under the effect of propylphosphonic anhydride (T3P). The previous amide coupling step of **5a** (R^2^ = 2-(methylamino)-2-oxoethyl) and **5b** (R^2^ = 2-pyridylmethyl) provided MOC-11 (R^1^ = *n-*pentyl) and MOC-33 (R^1^ = *n-*hexyl), respectively. The intermediate **5c** (R^2^ = diBoc-agmantyl, Boc = *tert*-butoxycarbonyl) [39] resulted in intermediates **7a**–**i**, which were directly subjected to acidic conditions for Boc deprotection to obtain the MOC final compounds 23, 24, 26, 27, 28, 29, 30, 31, and 32.

### 2.3. Comparative Binding Evaluation of MOC Compounds Using Differential Static Light Scattering (DSLS)

We studied the binding of MOC compounds with NS3 by differential static light scattering (DSLS) [40]. The NS3 domain stability upon binding to NS4A was measured by monitoring denatured protein aggregation upon increasing the temperature gradually from 25.0 to 85.0 °C (0.5 °C increments) at 600 nm. If a ligand binds to the protein, this increases a protein’s aggregation temperature (*T*_agg_) [40,41]. DSLS provides an advantage of being a label-free binding assay technique and Δ*T*_agg_ (of liganded protein−free protein) can be used to assess relative binding affinities of ligands [42]. For NS3, the Δ*T*_agg_ values that occurred upon binding of the MOC compounds were measured and compared to that of NS4A_21`–33`_ as a positive control. MOC-11 (Δ*T*_agg_ 0.82 °C 54.6% of the positive control) MOC-24 (Δ*T*_agg_ 0.60 °C 39.8%), and MOC-23 (Δ*T*_agg_ 0.32 °C 19.5%) demonstrated significant affinities towards NS3 (Figure 3). 

### 2.4. Fluorescence Anisotropy (FA) Competition Assay of MOC Compounds with NS4A 

This assay measures the potency by which synthetic peptidomimetics affect the binding of labeled NS4A_21`–33`_. Before testing the MOC compounds, the binding affinity of fluorescein isothiocyanate NS4A_21`–33`_ (FITC-NS4A_21`–33`_) was first determined (see Appendix A) based on our previous work [29]. For this particular protein batch, the optimal ratio to be used in the FA affinity test was calculated to be 0.5 µM NS3 and 0.1 µM FITC-NS4A_21`–33`_. Compounds that efficiently compete with NS4A_21`–33`_ are expected to decrease the fluorescence emitted from NS3/FITC-4A_21-34`_ mixture. The results showed that MOC-24 had the strongest competition potency in this assay as it showed a half maximal inhibitory concentration (IC_50_) of 1.9 µM (Table 1, Figure 4), followed by MOC-23 (4.7 µM). MOC-32 also showed moderate inhibition of NS4A binding (competition IC_50_ 7.7 µM). Compounds MOC-11, MOC-30, MOC-31, and MOC-33 demonstrated a weaker affinity at competition IC_50_ values between 12 and 40 µM, while MOC-26, MOC-27, MOC-28, and MOC-29 exerted no inhibition of NS4A binding up to 100 µM.

### 2.5. Enzyme Inhibition Assay

MOC-24 was selected for a qualitative testing for its inhibition of NS3 protease activity. Experiments were carried out using the SensoLyte^®^ 520 HCV Protease Assay Kit. This assay measures the recovered fluorescence emitted upon cleavage of the labeled peptide substrate representing the NS4A/NS4B junction [43]. Considering that NS3 without NS4A is almost a thousand-times less catalytic [23], the method was modified to suit the purpose of our allosteric inhibition assay. For instance, we used a large amount of free NS3 protein (2.00 µM) as a smaller concentration did not show measurable catalytic activity in the absence of cofactors. The addition of 4.00 µM amounts of MOC-24 resulted in a complete loss of NS3 catalytic activity. As a positive control, an equimolar mixture of NS4A with NS3 (equimolar, 2.00 µM each) caused only a 9-fold increase in the catalytic activity of the enzyme (Figure 5), which was far less than the reported value [23], possibly due to the large amount of protein used. 

### 2.6. Molecular Dynamics Simulation for MOC-24 and MOC-26

In order to rationalize the differences in the binding affinities of MOC-24 and MOC-26 (as representatives of active and inactive compounds, respectively), we performed a molecular dynamics (MD) simulation for 20.0 ns at 300 °K after manual docking of both compounds to place them in the intended pocket followed by energy relaxation. The MD simulation tries to disrupt non-bonding interactions between the ligand and the binding site residues by applying energy snapshots that result in conformational excitations followed by relaxations until equilibration occurs [44]. Equilibration of the system can be determined by plotting the protein’s backbone–ligand root mean square deviation (PL-RMSD) vs. the simulation time. If certain protein–ligand interactions stayed uninterrupted after the equilibrium was reached, this indicates a higher possibility of occurrence of the postulated binding mode [45].

First, an MD control experiment was performed on the NS3/4A_21`–32`_ crystal structure (PDB Code 1NS3). As expected, this viral peptide–protein complex maintained a high degree of stability and low residue fluctuations during most of the simulation period (see Appendix A). The MD simulation model of MOC-24 demonstrated three important features: (a) the ligand resided during the entire simulation period (20.0 ns) within the boundaries of the presumed binding region of NS4A_25`–28`_ located between the A0 and A1 β-strands; (b) the imidazole nucleus had low interaction fluctuations; and (c) the *n*-hexyl side chain stayed deep in the hydrophobic pocket accommodating the Ile-25` side chain (a video illustration is available as part of the Appendix A). Monitoring the PL-RMSD of the NS3/MOC-24 simulation indicated that the conformational fluctuations of the NS3 protein (backbone atoms) equilibrated after ~10.0 ns and this status was maintained towards the end of the simulation period (Figure 6). Meanwhile, the ligand (MOC-24) equilibrated at ~13.4 ns. Both the protein and the ligand clearly showed the least fluctuations after 17.5 ns and until the simulation ended at 20 ns, indicating that the complex reached stability at this binding mode. 

Accordingly, we looked carefully at the conformational behavior of the crucial imidazole-2,5-dicarboxamide core at the three phases of the ligand conformational dynamics. In general, the imidazole nucleus and especially its NH at position 1 showed low fluctuation (Figure 7). During the first phase (0–10.0 ns), we noticed that the imidazole-2,5-dicarboxamide core maintained two hydrogen bonds with Val-35 of the A1 β-strand, the presumed situation according to the design hypothesis described above (Figure 8A). Around 10 ns, the imidazole ring started to detect and interact with Gln-8, a residue within the opposite A0 β-strand located at the periphery of the NS3 protein (Figure 8B). At ~13.3 ns, the imidazole-2,5-dicarboxamide settled to embrace the Gln-8 backbone via two hydrogen bonding interactions and, in the process, lost contact with Val-35. After this major flip, the imidazole committed to Gln-8 and showed low fluctuation towards the end of the simulation (Figure 8C,D). 

The behavior of the hydrophobic substituent attached to the C2 carboxamide of MOC-24 was a critical issue in our model as this moiety of the compound is the only difference between the active MOC-24 and the inactive MOC-26. The *N*^2^-(*n*-hexyl) group of MOC-24 stayed in the deep hydrophobic pocket although it did not show strong commitments to a certain amino acid in the active site. In this regard, Leu-64 and Pro-88 were the most interacting residues with the *n*-hexyl side chain. This hydrophobic *n*-hexyl showed a low RMSF compared to the other polar *N*^5^-agmantyl substituent (Figure 7). The MD simulation results are consistent with lack of binding of the inactive MOC-26, which contains a bulkier hydrophobic substituent that ends with a cyclohexyl group.

As the simulation time progressed, this cyclohexyl group was forced outside the hydrophobic pocket. In fact, the whole MOC-26 molecule left the NS4A binding region located in between the A0 and A1 β-sheets. The imidazole made only minor contacts with Gln-8, while most of the ligand clearly popped outside to settle near the protein surface (Figure 9). This movement indicates a lack of affinity of MOC-26 for the NS4A binding pocket. The last point obtained from the MD study was about the role of the polar agmantyl group in MOC-24. This group served as a polar head suitable for the microenvironment around the Arg-28` of the bound NS4A. The group alternated interactions at the periphery during the simulation between Glu-30, Gln-28, and the solvent. This is not an odd observation as Arg-28` showed similar behavior in reported NS3/4A crystal structures [46,47,48] (see Appendix A). 

## 3. Discussion and Conclusions

The introduction of DAA combinations allows for broadening the efficacies of HCV treatment to the pan-genotypic orbit. These combinations are made possible by the identification of more targets in the viral proteome. For instance, Vosevi^®^ is a combination of sofosbuvir (NS5B inhibitor), velpatasvir (NS5A inhibitor), and voxilaprevir (NS3/4A protease inhibitor) that cover HCV genotypes 1–6. We were truly motivated by the inclusion of NS4A competitors as a less-studied target toward potential further emboldening of the arsenal of DAA combinations. However, in our extensive literature search, we could not find a precedent for designing and/or screening small-molecule peptide mimics of NS4A. An additional advantage of targeting NS4A is that this vital component is a versatile small peptide commonly expressed in the entire *Flaviviridae* family of viruses, which, besides HCV, includes other dangerous pathogenic viruses such as the Zika, West Nile, and dengue viruses. Therefore, effective NS4A peptidomimetics may constitute a possible general approach to combat *Flaviviridae.*


Our approach of the peptidomimetic design depends on simple observations: the conformation of the bound NS4A and the relevance of the residues involved in the design. The virtually designed 1*H*-imidazole-2,5-dicarboxamide appeared to satisfactorily reproduce the stereochemistry of these NS4A features. Indeed, we could experimentally confirm that such compounds display binding affinity towards NS3 genotype-4, the most abundant type in the Middle East [49,50]. Our work allowed quantifying the binding affinity of our designed MOC compounds. The FA competition assay confirmed their binding to the NS4A binding site between the A0 and A1 β-strands. Both DSLS and FA assays were generally in agreement, showing that MOC-11, MOC-23, and MOC-24 exerted good affinities towards NS3. 

Structurally, the three compounds contain *n-*pentyl or its homologue *n*-hexyl in the R^1^ position. Larger hydrophobic groups at this position (MOC-26 to MOC-31) had lower binding affinities in both DSLS and FA tests, indicating that the pocket has limited bulk tolerance. Regarding the R^2^ position, all screened compounds contained an agmantyl group, except for MOC-11 and MOC-33. The agmantyl group was chosen as a mimic of the NS4A Arg-28`. In addition, the agmantyl moiety was hypothesized to act as a hooking group and direct the imidazole scaffold to its intended position. MOC-11 and MOC-33 were synthesized appending two different polar groups at the R^2^ position. MOC-11 encompassed a glycinyl moiety (backbone mimic). 

Compared to the agmantyl analogue MOC-23, MOC-11 produced a higher increase in *T*_agg_, but showed lower competition potency in the FA assay. This indicates that the presence of the Arg-28` side chain may be replaced with another polar group. This structure–activity relationship (SAR) feature is important for future research towards optimization of the NS4A peptidomimetics by removing the guanidinyl moiety—a less desirable group in drug discovery, due to high ionizability and poor pharmacokinetics. MOC-33 presented a more radically different variant from the Arg-28` side chain group. This compound showed much weaker binding in both tests, indicating that a successful peptide mimic should include close mimics of those amino acid features (glycinyl or agmantyl).

MOC-24, a homologue of MOC-23, exerted confirmed binding activities as revealed in both DSLS and FA tests. It showed ~1/3 of the thermal stabilization (Δ*T*_agg_) of NS3 when compared to the natural cofactor NS4A_21`–33`_. In the FA assay, it demonstrated the highest ability to displace NS4A_21`–33`_ from NS3 binding among the tested compounds. Moreover, our in vitro protease assay showed that MOC-24 can inhibit NS3 by forming an inactive complex. However, this test could not be further developed to a quantitative application due to the very week catalytic activity of the NS3 in in vitro conditions. 

The MD simulation, conducted in the course of this study, was useful in validating the design hypothesis and in explaining the binding potency differences between MOC-24 and MOC-26 (as representatives for active and inactive compounds, respectively). It was remarkable that MOC-24 stayed in the region between A0 and A1 β-sheets and the *n*-hexyl group completely buried deep in a hydrophobic pocket for the entire MD simulation time. A question that frequently comes to mind is what conformational effects occurred consequently to binding of NS4A peptidomimetics (e.g., MOC-24). This question should be of secondary importance, anyway, as prevention of NS4A binding by non-peptide mimics should lead to the inability of NS3 to perform its catalytic function. 

Collectively, our work introduces a new strategy to interfere with NS3 protease activity using compounds that represent non-peptide mimics of an NS4A fragment (Ile-25` to Arg-28`). We must acknowledge that our compounds represented only four amino acids at the core of NS4A and covered a limited region of the binding site. Considering their promising activities and limited molecular weight (MOC-24, MW 351.5), the results with imidazole-2,5-carboxamides demonstrate that NS4A binding is a good target for the discovery of a newer generation of DAAs. Our work can be considered as a bridge between the peptide and non-peptide mimics of NS4A. Further expansion upon this lead can pave the way for the future discovery of NS4A competitors as clinical candidates.

## 4. Experimental

### 4.1. Chemical Synthesis

Solvents and reagents were purchased from Sigma-Aldrich (St. Louis, MO, USA), VWR (Radnor, PA, USA), or Alfa Aesar (Heffrey, MA, UK). When needed, solvents were dried according the procedures described in the literature. Unless otherwise stated, the reactions were performed under an inert atmosphere of nitrogen. Microwave (MW) reactions were performed using Milestone StartSynth™ reactor (Milestone Inc., Sorisole, Italy). Melting points (mp) were determined in open capillary tubes using electrothermal apparatus (Stuart, Staffordshire, UK) and are uncorrected. NMR results were recorded on Bruker DPX-300 MHz (Bruker, Fällanden, Switzerland). HPLC–MS was performed on an Agilent 1100/ZQ MSD including a C18 column and a diode-array UV detector. The mobile phase (containing 0.01 M ammonium acetate) was a gradient starting from 20% acetonitrile/80% water to 80% acetonitrile/20% water. Purities are reported according to percentage of peak areas at wavelength 254 nm. According to LC–MS analyses, all compounds in this study were confirmed to have 95% purity or higher. Infrared spectra were recorded on a Thermo Scientific Nicolet iS10 Fourier transform (FT)-IR Spectrometer. In this report, we only listed the important IR stretching bands, including NH, OH, CH, C=O, C=N, and/or C=C. In FT-IR, all samples were measured neat. The synthesis of intermediates 1,2-bis(*tert*-butoxycarbonyl)-3-(4-aminobutyl)guanidine [39], ethyl (*Z*)-3-amino-3-(hydroxyimino)propanoate (**2**) [38], and 5-(*tert*-butyl) 2-ethyl 1H-imidazole-2,5-dicarboxylate (**3**) [36,37] are described below and their characterization data were found to be in agreement with the literature data.

#### 4.1.1. Synthesis of 5-(Tert-Butyl) 2-Ethyl 1H-Imidazole-2,5-Dicarboxylate (**3**)

Water (0.6 mL/mmol) was added dropwise over a period of 2 h to a stirred mixture of ethyl cyanoformate **1** (1.0 equiv., 3 g, 30.3 mmol), hydroxylamine hydrochloride (1.5 equiv., 3.16 g, 45.4 mmol), and sodium carbonate (0.77 equiv., 2.470 g, 23.3 mmol) in ethanol (24.0 mL) at room temperature (rt). Upon completion, the reaction was quenched, and the solvent was removed under a vacuum. The resulting residue was extracted with dichloromethane, and the combined organic layers were washed with brine, dried over MgSO_4_, filtered, and concentrated to afford a white solid of the intermediate (*Z*)-2-amino-2-(hydroxyimino) acetate (**2**). The intermediate **2** was recrystallized from chloroform and *n*-heptane to afford white crystals (1.881 g, 47%), mp 59–61 °C, which were used without further purification for the next step. This intermediate (1.0 equiv., 1.320 g, 10 mmol) and *tert-*butyl propiolate (1.0 equiv., 1.262 g, 10 mmol) and Et_3_N (1.0 equiv., 1.40 mL, 1 mmol) were added to toluene (5.50 mL). The resulting solution was microwave-irradiated at 120 °C for 10 min at 300 W, after a heating ramp, for 3 min. The solvent was evaporated, and the final product was purified by crystallization from ethyl acetate (EtOAc)/hexane to obtain 128 mg (53.3%) of **3**, mp 160–165 °C. ^1^H NMR (300 MHz, DMSO DMSO-*d*_6_) δ_H_ ppm 13.72 (br., 1 H), 7.87 (s, 1 H), 4.34 (q, *J* = 7.2 Hz, 2 H), 1.51 (s, 9 H), 1.32 (t, *J* = 7.0 Hz, 3 H); ^13^C NMR (75 MHz, CDCl_3_) δ_c_ 169.7, 136.5, 129.5, 129.1, 128.9, 127.8, 123.8, 29.7; IR (FT-IR, cm^−1^): 3025.3, 2916.8, 2843.6, 1695.0, 1619.3, 1594.0, 1571.3, 1505.7; LC–MS (ESI), RT = 1.41 min, *m*/*z* 241.3 [M + H]^+^.

#### 4.1.2. Synthesis of 2-(Ethoxycarbonyl)-1H-Imidazole-4-Carboxylic Acid (**4**)

A solution of 5-*tert*-butyl 2-ethyl 1H-imidazole-2,5-dicarboxylate (1 equiv., 4.29 g, 17.6 mmol) and TFA (21.1 mL) in dichloromethane (DCM, 14.0 mL) was stirred for 12 h at room temperature (rt). The solvent was evaporated under a vacuum and the remaining residue was treated with MeOH (70.0 mL) and filtered to give **4** as a yellow solid (2.841 g, 86.29%), mp > 230 °C (decomposed). ^1^H NMR (300 MHz, DMSO–DMSO-*d*_6_) δ_H_ ppm 13.72 (br. s, 1 H); 12.59 (br. s, 1 H), 7.91 (s, 1H), 4.33 (q, *J =* 6.9 Hz, 2H), 1.32 (t, *J =* 7.0 Hz, 3H); LC–MS (ESI), RT = 1.55 min, *m*/*z* 185.3 [M + H]^+^.

#### 4.1.3. General Procedure for the Synthesis of Ethyl 4-(N-Substituted Carbamoyl)-1H-Imidazole-2-Carboxylate (**5a**‒**5c**)

Under an inert atmosphere, carboxylic acid (1.0 equiv., 5.21 mmol) was dissolved in dry THF (26 mL) then EDCI (1.0 equiv., 0.998 g, 5.21 mmol), HOBt (1.0 equiv., 0.704 mg, 5.21 mmol), diisopropylethylamine (DIPEA) (1.5 equiv., 1.35 mL, 7.80 mmol), and the amine R^2^NH_2_ (1.2 equiv., 6.25 mmol) were added in the indicated order. The mixture was stirred at rt for 1 h then heated to 60 °C. After 3 h, EDCI (0.5 equiv., 0.500 g, 2.60 mmol), HOBt (0.5 equiv., 0.352 g, 2.60 mmol), and DIPEA (0.75 equiv., 0.677 mL, 3.90 mmol) were added to the mixture. After completion, the mixture was purified by column chromatography using gradient elution (50–100% EtOAc in cyclohexane).


*Ethyl 4-((2-(methylamino)-2-oxoethyl)carbamoyl)-1H-imidazole-2-carboxylate (*
**5a**
*)*


The yield was 77%, mp 74–76 °C. ^1^H NMR (300 MHz, DMSO-*d*_6_) δ_H_ ppm 8.19 (br. s., 1H), 7.81 (s, 2H), 4.35 (q, *J =* 7.1 Hz, 2H), 3.81 (d, *J =* 5.6 Hz, 2H), 2.59 (d, *J =* 4.5 Hz, 3H), 1.33 (t, *J =* 7.1 Hz, 3H); LC–MS (ESI), RT = 1.57 min, *m/z* 255.3 [M + H]^+^.


*Ethyl 5-((pyridin-2-ylmethyl)carbamoyl)-1H-imidazole-2-carboxylate (*
**5b**
*)*


The intermediate **5b** was obtained as an oily substance and its yield was 94%. ^1^H NMR (300 MHz, DMSO-*d*_6_) δ_H_ ppm 13.24 (br. s. 1 H), 8.69 (t, *J =* 5.6 Hz, 1H), 8.51 (d, *J =* 4.1 Hz, 1H), 8.41 (s, 1H), 7.81 (s, 1H), 7.74 (t, *J =* 7.0 Hz, 1H), 7.19‒7.35 (m, 2H), 4.53 (d, *J =* 6.0 Hz, 2H), 4.34 (q, *J =* 7.1 Hz, 2H), 1.32 (t, *J =* 7.0 Hz, 3H); LC–MS (ESI), RT = 1.14 min, *m/z* 275.3 [M + H]^+^.


*Ethyl (E)-4-((4-(2,3-bis(tert-butoxycarbonyl)guanidino)butyl)carbamoyl)-1H-imidazole-2-carboxylate (*
**5c**
*)*


To a solution of 1,4-diaminobutane (2.0 equiv., 0.303 g, 3.44 mmol) in THF (2.3 mL) was added a solution of 1,3-bis(*tert*-butoxycarbonyl)-2-methyl-2-thiopseudourea (1.0 equiv., 500 mg, 1.72 mmol) in THF (1.7 mL/mmol) within 0.5 h. The solution was stirred at room temperature for 1 h. After completion, the solvent was removed under vacuum and the product was purified by column chromatography on a silica gel using a mixture of DCM/MeOH to give [[(4-aminobutyl)amino](carboxyamino)methylene]carbamic acid di-*tert*-butyl ester (0.250 g, 52.9%); ^1^H NMR (300 MHz, CDCl_3_) δ_H_ ppm 11.51 (br. s., 1 H), 8.36 (br. s., 1 H), 3.40‒3.52 (m, 2 H), 2.75 (t, *J* = 6.7 Hz, 2 H), 1.51‒1.71 (m, 4 H), 1.52 (s, 9 H), 1.51 (s, 9 H); LC–MS (ESI), RT = 2.44 min, *m/z* 331.8 [M + H]^+^_._ The produced amine [[(4-aminobutyl)amino](carboxyamino)methylene]carbamic acid di-*tert*-butyl ester was reacted with the carboxylic acid **4** according to the general procedure for amide coupling as described above (Section 4.1.3). The intermediate **5c** was collected as an oily substance; the yield was 0.097 g, 36.2%. The compound was used in the next step, the same day, without further characterization.

#### 4.1.4. General Procedure for Synthesis of **6a**–**6c**

The ethyl esters (1.0 equiv.) were dissolved in THF/water 4:1 (4 mL/mmol) then LiOH (3.0 equiv.) was added. The mixture was stirred at rt until completion. Afterward, the solvent was evaporated under a vacuum. The remaining residues was resuspended in EtOH/toluene and evaporated. No further purification or characterization was performed, and the salts were used for the next step.

##### N^5^-(2-(Methylamino)-2-oxoethyl)-N^2^-(n-pentyl)-1H-imidazole-2,5-dicarboxamide (MOC-11)

Under an inert atmosphere, the lithium salt of **6a** (180 mg, 0.78 mmol) was dissolved in dry THF (2.3 mL, 3 mL/mmol) then propanephosphonic acid anhydride (1.17 mL of 50% solution in THF, 3.0 equiv.) and the *n*-pentylamine (0.11 mL, 1.2 equiv.) were added. The mixture was stirred at room temperature for 72 h. After 48 h, additional 1.0 equiv. of T3P was added to the mixture. Upon reaction completion, the solvent was evaporated under a vacuum, and the residue was purified on a silica gel column with gradient elution 0%–20% MeOH in CHCl_3_ to give 0.107 g (46.0%) of MOC-11 as a white solid, mp 87 °C. ^1^H NMR (300 MHz, methanol-*d*_4_) δ_H_ ppm 7.76 (s, 1H), 4.04 (s, 1H), 3.74 (t, *J =* 6.4 Hz, 2H), 3.38‒3.51 (m, 1H), 2.78 (s, 2H), 1.80‒2.00 (m, 2H), 1.53‒1.80 (m, 2H), 1.20‒1.53 (m, 3H), 0.86‒1.07 (m, 3H); ^13^C NMR (75 MHz, methanol-*d*_4_) δ_C_ ppm 170.8, 48.5, 48.2, 47.9, 47.6, 47.4, 47.1, 46.8, 41.8, 38.9, 29.0, 28.9, 25.0, 22.1, 13.0; LC–MS (ESI), RT = 1.46 min, m/z 296.4 [M + H]^+^.

##### N^2^-(n-Hexyl)-N^5^-(pyridin-2-ylmethyl)-1H-imidazole-2,5-dicarboxamide (MOC-33)

This compound was prepared according to the procedure described above for the preparation of **MOC-11** starting from **6b** and *n*-hexylamine to obtain MOC-33 as an oily substance. The yield was 33%. ^1^H NMR (300 MHz, methanol-*d*_4_) δ_H_ ppm 8.52 (d, *J =* 4.5 Hz, 1H), 7.70‒7.93 (m, 2H), 7.45 (d, *J =* 7.9 Hz, 1H), 7.19‒7.40 (m, 1H), 4.71 (s, 2H), 3.39 (t, *J =* 7.0 Hz, 2H), 1.51‒1.76 (m, 2H), 1.37 (br. s., 6H), 0.93 (t, *J =* 6.4 Hz, 3H); ^13^C NMR (75 MHz, methanol-*d*_4_) δ_C_ ppm 158.6, 157.8, 148.5, 137.5, 122.5, 121.9, 121.6, 48.5, 48.2, 47.9, 47.7, 47.4, 47.1, 46.8, 43.7, 39.0, 31.3, 29.2, 26.3, 22.3; LC–MS (ESI), RT = 2.74 min, *m/z* 330.3 [M + H]^+^.

##### N^5^-(4-Guanidinobutyl)-N^2^-(n-pentyl)-1H-imidazole-2,5-dicarboxamide (MOC-23)

The lithium carboxylate salt of **6c** was reacted with *n-*pentylamine according the procedure described above for the preparation of MOC-11. The collected intermediate *N^5^*-[[4-(2,3-bis(*tert*-butoxycarbonyl)guanidino)butyl]carbamoyl]-*N^2^*-(*n*-pentyl)-*1H*-imidazole-2,5-dicarboxamide (**7a**) was subjected to a deprotection reaction for the removal of Boc groups. Thus, it was dissolved in dry DCM (5 mL/mmol), hydrochloric acid (in 1,4-dioxane solution (5 mL/mmol) was added, and the reaction mixture was stirred at rt for 2 h. After completion, the reaction mixture was concentrated and dried under a vacuum and purified using preparative HPLC to furnish MOC-23 (27.3%), mp 106 °C. ^1^H NMR (300 MHz, methanol-*d*_4_) δ_H_ ppm 7.83 (s, 1H), 3.36‒3.60 (m, 5H), 3.25 (br. s., 3H), 1.69 (br. s., 7H), 1.17‒1.50 (m, 6H), 0.95 (br. s., 4H); ^13^C NMR (75 MHz, methanol-*d*_4_) δ_C_ ppm 48.5, 48.3, 48.0, 47.7, 47.4, 47.1, 46.8, 40.8, 39.3, 28.9, 28.7, 25.9, 22.1, 13.0; LC–MS (ESI), RT = 1.90 min, *m/z* 338.4 [M + H]^+^.

##### N^5^-(4-Guanidinobutyl)-N^2^-(n-hexyl)-1H-imidazole-2,5-dicarboxamide (MOC-24)

This compound was prepared according to the procedure described above for the synthesis of MOC-23. The product was obtained as a yellowish oily substance (37.1%). ^1^H NMR (300 MHz, methanol-*d*_4_) δ_H_ ppm 7.81 (br. s., 1H), 3.39‒3.57 (m, 4H), 3.07‒3.27 (m, 3H), 1.61‒1.84 (m, 8H), 1.55 (t, *J =* 7.1 Hz, 1H), 1.18‒1.48 (m, 9H), 0.93 (br. s., 4H); ^13^C NMR (75 MHz, methanol-*d*_4_) δ_C_ ppm 48.5, 48.2, 48.0, 47.7, 47.4, 47.1, 46.8, 40.8, 39.2, 31.3, 26.4, 25.9, 22.3, 13.0; LC–MS (ESI), RT = 1.78 min, *m/z* 352.3 [M + H]^+^.

##### N^2^-(2-(Cyclohexyloxy)ethyl)-N^5^-(4-guanidinobutyl)-1H-imidazole-2,5-dicarboxamide (MOC-26)

This compound was prepared according to the procedure described above for the synthesis of MOC-23. The product was obtained as a yellowish oily substance (19.4%). ^1^H NMR (300 MHz, methanol-*d*_4_) δ_H_ ppm 7.76 (br. s., 1H), 3.50‒3.75 (m, 7H), 3.44 (br. s., 3H), 3.25 (br. s., 3H), 1.81‒2.09 (m, 5H), 1.63‒1.81 (m, 9H), 1.45‒1.63 (m, 2H), 1.11‒1.44 (m, 7H); ^13^C NMR (75 MHz, methanol-*d*_4_) δ_C_ ppm 77.5, 65.6, 48.5, 48.2, 47.9, 47.6, 47.4, 47.1, 46.8, 40.7, 37.0, 31.9, 25.8, 25.6, 23.7; LC–MS (ESI), RT = 2.46 min, *m/z* 213.2 [M + H]^+^

##### N^2^-(3-(Cyclohexyloxy)propyl)-N^5^-(4-guanidinobutyl)-1H-imidazole-2,5-dicarboxamide (MOC-27)

This compound was prepared according to the procedure described above for the synthesis of MOC-23. The product was obtained as a yellowish oily substance (41.8%). ^1^H NMR (300 MHz, methanol-*d*_4_) δ_H_ ppm 7.73 (s, 1H), 3.38‒3.76 (m, 7H), 3.13‒3.30 (m, 3H), 1.79‒2.07 (m, 5H), 1.63‒1.79 (m, 7H), 1.56 (d, *J =* 7.5 Hz, 1H), 1.12‒1.42 (m, 5H); ^13^C NMR (75 MHz, methanol-*d*_4_) δ_C_ ppm 77.5, 65.6, 48.5, 48.2, 47.9, 47.6, 47.4, 47.1, 46.8, 40.7, 37.0, 31.9, 25.8, 25.6, 23.7; LC–MS (ESI), RT = 1.98 min, *m/z* 408.3 [M + H]^+^.

##### N^5^-(4-Guanidinobutyl)-N^2^-(2-phenoxyethyl)-1H-imidazole-2,5-dicarboxamide (MOC-28)

This compound was prepared according to the procedure described above for the synthesis of MOC-23. The product was obtained as an off-white solid; mp 137–139 °C (47.8%). ^1^H NMR (300 MHz, methanol-*d*_4_) δ_H_ ppm 8.01 (s, 1H), 7.28 (t, *J =* 7.9 Hz, 2H), 6.82‒7.10 (m, 3H), 4.19 (t, *J =* 5.1 Hz, 2H), 3.84 (t, *J =* 5.1 Hz, 2H), 3.44 (br. s., 2H), 3.25 (br. s., 2H), 1.69 (br. s., 5H); ^13^C NMR (75 MHz, methanol-*d*_4_) δ_C_ ppm 129.2, 114.2, 48.5, 48.2, 47.9, 47.6, 47.4, 47.1, 46.8; LC–MS (ESI), RT = 1.86 min, *m/z* 388.4 [M + H]^+^.

##### N^5^-(4-Guanidinobutyl)-N^2^-(3-phenoxypropyl)-1H-imidazole-2,5-dicarboxamide (MOC-29)

This compound was prepared according to the procedure described above for the synthesis of MOC-23. The product was obtained as an off-white solid mp 135–138 °C (37.1%). ^1^H NMR (300 MHz, methanol-*d*_4_) δ_H_ ppm 7.73 (s, 1H), 7.13‒7.38 (m, 2H), 6.80‒7.05 (m, 3H), 4.10 (t, *J =* 5.9 Hz, 2H), 3.61 (t, *J =* 6.7 Hz, 2H), 3.44 (br. s., 2H), 3.14‒3.30 (m, 2H), 2.11 (quin, *J =* 6.3 Hz, 2H), 1.52‒1.84 (m, 5H); ^13^C NMR (75 MHz, methanol-*d*_4_) δ_C_ ppm 129.1, 120.4, 114.2, 65.3, 48.5, 48.2, 47.9, 47.6, 47.4, 47.1, 46.8, 40.8, 37.9, 36.4, 29.0, 26.6, 25.8; LC–MS (ESI), RT = 1.51 min, *m/z* 402.3 [M + H]^+^

##### N^5^-(4-Guanidinobutyl)-N^2^-(4-isopropoxybutyl)-1H-imidazole-2,5-dicarboxamide (MOC-30)

This compound was prepared according to the procedure described above for the synthesis of MOC-23. The product was obtained as an oily substance (38.4%). ^1^H NMR (300 MHz, methanol-*d*_4_) δ_H_ ppm 7.73 (s, 1H), 3.35‒3.74 (m, 8H), 3.14‒3.28 (m, 2H), 1.69 (br. s., 9H), 1.16 (d, *J =* 6.2 Hz, 6H); ^13^C NMR (75 MHz, methanol-*d*_4_) δ_C_ ppm 71.5, 67.4, 48.5, 48.2, 47.9, 47.6, 47.4, 47.1, 46.8, 40.8, 38.7, 37.9, 27.1, 26.6, 26.1, 25.8, 21.0; LC–MS (ESI), RT = 2.08 min, *m/z* 382.3 [M + H]^+^.

##### N^5^-(4-Guanidinobutyl)-N^2^-(6-methylheptyl)-1H-imidazole-2,5-dicarboxamide (MOC-31)

This compound was prepared according to the procedure described above for the synthesis of MOC-23. The product was obtained as a solid, mp 83–84 °C (23.7%). ^1^H NMR (300 MHz, methanol-*d*_4_) δ_H_ ppm 7.74 (s, 1H), 3.38‒3.57 (m, 2H), 3.14‒3.30 (m, 2H), 1.54‒1.83 (m, 6H), 1.22‒1.54 (m, 9H), 0.81‒1.09 (m, 7H); ^13^C NMR (75 MHz, methanol-*d*_4_) δ_C_ ppm 48.5, 48.2, 47.9, 47.6, 47.3, 47.1, 46.8, 40.8, 39.5, 30.6, 28.6, 26.7, 25.8, 23.8, 22.7, 13.0, 9.8; LC–MS (ESI), RT = 1.19 min, *m/z* 380.3 [M + H]^+^.

##### N^5^-(4-guanidinobutyl)-N^2^-(3-methylpentyl)-1H-imidazole-2,5-dicarboxamide (MOC-32)

This compound was prepared according to the procedure described above for the synthesis of MOC-23. The product was obtained as an off-white oily substance (41.2%). ^1^H NMR (300 MHz, methanol-*d*_4_) δ_H_ ppm 7.73 (s, 1H), 3.35‒3.58 (m, 4H), 3.14‒3.30 (m, 2H), 1.69 (br. s., 6H), 1.35‒1.59 (m, 3H), 1.24 (td, *J =* 7.0, 13.7 Hz, 1H), 0.77‒1.09 (m, 6H); ^13^C NMR (75 MHz, methanol-*d*_4_) δ_C_ ppm 48.5, 48.2, 47.9, 47.6, 47.4, 47.1, 46.8, 40.8, 37.1, 35.9, 32.1, 29.1, 26.6, 25.8, 18.0, 10.2; LC–MS (ESI), RT = 1.79 min, *m/z* 352.3 [M + H]^+^.

### 4.2. Biological Screening

All reagents used in the biological screenings were purchased from Merck KGaA (Darmstadt, Germany) in molecular biology grade unless stated otherwise. 

#### 4.2.1. NS3 Protein

##### NS3 Constructs

A synthetic gene coding for the HCV NS3 domain of genotype 4a, the most abundant HCV in Saudi Arabia and Egypt [51], was synthesized by GenScript (Hong Kong, China), the nucleotide sequence was optimized for *Escherichia. coli* codon usage. The synthetic gene was cloned as a *Nde*I–*Bam*HI fragment into the expression vector pET-3a Novagen^®^. The obtained construct was sequenced to confirm the identity and frame of the expression construct. 

##### NS3 Protein Information

Accession GU085486.1

HCV genotype 4a (The most common genotype in Saudi Arabia) [52]

NS3 from 4 to 182 aa (L/E, F/E, I/Q, V/E, L/Q, C/S)

NS4A 632 to 685 aa (i/n) are:

G SVVIVGRVNL SGDTAYAQQT RGEESTQETS QTGRDTNENC GEVQVLSTAT QSFLGTAVNG VMWTVYHGAG SKTISGPKGP VNQMYTNVDQ DLVGWPSPPG VKSLTPCTCG ASDLYLVTRH ADVVPVRRRG DTRGALLSPR PISTLKGSSG GPLLCPMGHA AGLFRAAVST RGVAKAVDFV PVESLETT MRSP

NS4A/NS3 fusion protein expression in pET-28a

NS3 protease domain 1–181 aa plus an N-terminal T7 tag and C-terminal His tag (marked in uppercase)

M ASMTGGQQMG APITAYAQQT RGLFSTIVTS LTGRDTNENC GEVQVLSTAT QSFLGTAVNG VMWTVYHGAG SKTISGPKGP VNQMYTNVDQ DLVGWPSPPG VKSLTPCTCG ASDLYLVTRH ADVVPVRRRG DTRGALLSPR PISTLKGSSG GPLLCPMGHA AGLFRAAVCT RGVAKAVDFV PVESLETTMR sGSHHHHHH

These sequences were expressed in pET-3a

##### Protein Expression

The sequence of NS3 domain for genotype 4A, was expressed in *E. coli* Rosette (DE3) pLysS according to a standard protocol [14]. Therefore, a synthetic gene for NS3 domain was subcloned in the expression vector pET-3a. A 100 mL bacterial culture in Luria Broth (LB) medium grown overnight at 37 °C was used for inoculation of 10 L LB in a 14 L fermentation flask (New Brunswick Scientific Co., Enfield, CT, USA). The media was supplemented with ampicillin at 50 μg/mL. The culture was grown until the OD_600_ reached 0.5–0.6; it was then cooled to 25 °C and 1 mM isopropyl-β-thiogalactoside (IPTG) was added for expression overnight, followed by harvesting of cells. 

##### Protein Purification 

The produced protein was purified using equilibrated nickel nitrilotriacetic acid (Ni-NTA) beads and without removal of the poly-histidine tag. For this process, cells were first resuspended (1.00 g/5 mL) in buffer (50 mM N-2-hydroxyethyl-piperazine-N′-2-ethanesulfonic acid (HEPES), 0.300 M NaCl, 10.0% glycerol, 2.00 mM β-mercaptoethanol, pH 8.0). Lysozyme was added (1.00 mg/mL) followed by a protease inhibitor cocktail tablet and the suspension was sonicated. The cell lysate was centrifuged to collect the clear supernatant that contained the desired NS3 protein. The protein was purified using pre-equilibrated Ni-NTA beads (Qiagen, Germantown, MD, USA). The beads were washed with buffer (50.0 mM HEPES, 0.300 M NaCl, 10.0% glycerol, 2.00 mM β-mercaptoethanol, 20.0 mM imidazole, pH 8.0) and eluted with another buffer (50.0 mM HEPES, 0.300 M NaCl, 10.0% glycerol, 2.00 mM β-mercaptoethanol, 350 mM imidazole, pH 8.0). The fractions were collected and concentrated using an Amicon Ultra-4 3000 MWCO centrifugal device (Merck KGaA (Darmstadt, Germany). The protein purity after Ni-affinity purification step was not less than 70.0%. The purity, as estimated by Sodium Dodecyl Sulfate Polyacrylamide Gel Electrophoresis (SDS-PAGE), was sufficient to perform all investigations of this study and the protein was stable for several hours at test conditions [51]. The concentration of NS3 in the final concentrate was measured using a Nanodrop™ nanoscale spectrophotometer.

When needed, further purification of the protein was accomplished on a Superdex 75 16/90 column (GE Healthcare, Chicago, IL, USA) equilibrated in 20.0 mM HEPES, 10 mM DDT, 200 mM NaCl, pH 7.60 run at rate of 1 mL/min followed by SDS-PAGE for purity estimation.

#### 4.2.2. NS4A

The cofactor NS4A and the fluorescent fluorescein isothiocyanate NS4A (FITC-NS4A) were purchased from GenScript (Hong Kong, China). The NS4A structure was identical to that of the HCV genotype 4a with two lysine residues added at both the N- and C-termini. Thus, the structure of NS4A used in this study was LL-G_21_SVVIVGRIVLSG_33_-LL.

We studied the binding of NS4A and its mutants with NS3 by DSLS using Stargazer-2™ (Harbinger Biotechnology and Engineering Corporation, Toronto, Canada). This method assesses protein stability by monitoring aggregate formation using controlled, gradually elevated temperatures [40]. The NS3 domain stability upon binding to NS4A was measured by monitoring denatured protein aggregation upon increasing the temperature from 25.0 to 85.0 °C (0.5 °C increments) at 600 nm. The *T*_agg_ values were calculated automatically from raw data using Stargazer-AIR^®^ software. All experiments were repeated three times using the same batch of NS3 protein. 

#### 4.2.3. DSLS Binding Test

The NS3 domain (15.0 µM), alone or mixed with an equimolar equivalent of tested MOC derivative, was added to a binding buffer (20.0 mM HEPES, 10.0 mM DTT, 200 mM NaCl, pH 7.60) to a final volume 100 µL. The mixture was incubated at room temperature with gentle shaking for 2 h. Afterwards, 10 µL of the mixture was transferred into a clear bottomed Nunc 384-well plate and covered by 10 µL paraffin oil to minimize evaporation. Protein aggregation was monitored by tracking the change in scattered light that was detected by a charged coupled device (CCD) camera. Snapshot images of the plate were taken every 0.5 °C. The pixel intensities in a preselected region of each well were integrated using image analysis software to generate a value representative of the total amount of scattered light in that region. These intensities were then plotted against temperature for each sample well and fitted to obtain the aggregation temperature (*T*_agg_). Aggregation was monitored and analyzed to assess the effect of NS4A and its synthetic analogues on the stability of the NS3 as an indicator of binding. Each experiment was repeated three times. Statistical analysis was performed using GraphPad Prism v. 8.0^®^ and Instat^®^ v. 3.10 software.

#### 4.2.4. Binding and Competition Assay by Fluorescence Anisotropy

In a 96-well plate, we added a place binding buffer (20 mM HEPES, 10 mM DTT, 200 mM NaCl, pH 7.60) and a mixture of NS3 (0.40 µM) and FITC-NS4A (0.10 µM) at the calculated affinity constant concentration. A serial dilution of MOC compounds (dissolved in 1.50% DMSO in binding buffer) (1/2 dilution starting from 150 to 0.292 µM) were mixed gently for 60 min in a dark place and the fluorescence was measured at excitation/emission wavelengths of 480/520 nm. The competition IC_50_ was calculated according to the recommended equation embedded in GraphPad Prism v. 8.0^®^ and Instat^®^ software.

#### 4.2.5. Enzyme Inhibition Assay

The assay was performed using SensoLyte-520^®^ HCV protease assay kit fluorometric* (Anaspec, Fremont, CA, USA) according to a modified procedure to suit the purpose of determination of allosteric inhibition. NS3 (2.00 µM) was mixed with MOC-24 (4.00 µM) for 15.0 min. Afterwards, 5-FAM/QXL™520 fluorescence resonance energy transfer (FRET) peptide was added as instructed by the assay kit manual. The sequence of this FRET peptide (5-FAM-SLGRKIQIQ) was derived from the cleavage site of NS4A/NS4B. In the FRET peptide, the fluorescence of 5-FAM is quenched by QXL™520. Upon cleavage into two separate fragments by HCV NS3/4A protease, the fluorescence of 5-FAM is recovered, and can be monitored at 490 nm/520 nm (excitation/emission). Controls included buffer, compound, compound + NS3, NS3 + NS4A, and the FRET peptide separately. All test wells and controls were repeated in triplicate on the same 96-well plate.

### 4.3. Molecular Modeling

#### 4.3.1. Hardware and Software

The molecular modeling was performed on HP OMEN-X PC (Processor: Intel^®^ Core™ i9-7920x CPU @ 2.9 GHz, Installed Memory (RAM): 32.0 GHz, Operating Systems: Windows 10 Enterprise and Enterprise Linux 7 (Distribution CentOS-7). The molecular computational works were performed using Schrӧdinger Maestro™ Suite (Schrӧdinger, New York, NY, USA). 

#### 4.3.2. Protein and Ligand Preparation

The NS3/4A (non-fusion protein) crystal structure (PDB Code: 1NS3) was downloaded from the Protein Data Bank website (rcsb.org). The protein was prepared according to the standard procedure in the Protein Preparation module of Maestro [53]. Briefly, the first stage included Preprocess followed by simplification of the protein to a monomer, removal of unused ligands and water. Finally an energy minimization was performed using an OPLS3e force field [54]. MOC compounds were sketched using ACD/ChemSketch freeware (Advanced Chemistry Development, Inc., Toronto, ON, Canada), saved as an sdf file, and imported to Maestro. Ligands were optimized using LigPrep standard procedure embedded in the Maestro suite. The ligands selected for molecular dynamics were merged (by 3D visual guidance) onto the prepared NS3 protein after the removal of the co-crystallized NS4A_21`-32`_ peptide to form the complexes that entered in the next step. 

#### 4.3.3. Molecular Dynamics

Using Desmond’s System Builder module, the protein–ligand complex was prepared for MD simulation by applying an OPLS3e molecular mechanics force field after solvation in a water box (shape, orthorhombic; size calculation method, buffer; distance from protein, 10 × 10 × 10 Å; box volume, minimize volume) that contained 0.15 M NaCl as added ions. Other parameters were used as default in the System Builder. 

The resulting solvated ligand–protein complex was subjected to Desmond molecular dynamics simulation after setting the following parameters: simulation time, 20 ns; recording internal: trajectory, 10 ps; approximate number of frames, 1000; ensemble class, normal pressure and temperature (NPT); temperature, 300 K, pressure, 1.013 bar. Before starting the MD job, the system was allowed to relax using the default relaxation protocol. Other parameters were used as default.

## 5. Patents

El-Araby, M. E.; Omar, A. M.; El-Faky, M. A.; Soror, S. H.; Khayat, M. T.; Asfour, H. Z.; Bamane, F. H. Imidazole-based Compounds as hepatitis C virus inhibitors, US Application 16/384,472, 16 April 2019.

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
