# Peer review of "1H-Imidazole-2,5-Dicarboxamides as NS4A Peptidomimetics: Identification of a New Approach to Inhibit HCV-NS3 Protease"

_biomolecules, 2020, doi:10.3390/biom10030479_

Round 1

Reviewer 1 Report

This manuscript by Omar et al, seeks to find new inhibitors for HCV NS3 protein. Hepatitis C is a  huge problem, especially in the developing world. New treatments in the past decade are not accessible to poorer countries. The approach they used is clever and the results show moderate inhibition of NS3 if NS4 is used as the standard inhibitor. The presentation is clear even for a general audience. 

The main issue with the manuscript is that all experiments were done in-vitro. This makes it difficult to guage the significance of the findings. For instance how will mammalian cells, especially hepatic cell lines do when exposed to MOC 24? At a minimum, the authors should expose MOC 24 and 11 to mammalian cells to determine its toxicity. That will make the findings more relevant to readers. 

Author Response

This manuscript by Omar et al, seeks to find new inhibitors for HCV NS3 protein. Hepatitis C is a  huge problem, especially in the developing world. New treatments in the past decade are not accessible to poorer countries. The approach they used is clever and the results show moderate inhibition of NS3 if NS4 is used as the standard inhibitor. The presentation is clear even for a general audience.

Answer: Thanks very much.

The main issue with the manuscript is that all experiments were done in-vitro. This makes it difficult to guage the significance of the findings. For instance how will mammalian cells, especially hepatic cell lines do when exposed to MOC 24? At a minimum, the authors should expose MOC 24 and 11 to mammalian cells to determine its toxicity. That will make the findings more relevant to readers.

Answer:

Thank you for the comment. It should be considered as an imperative when we move from the proof of concept (this manuscript) towards application research. Nonetheless, we truly wanted to bring HHL hepatic cell lines described in the following paper:

"Clayton, Reginald F., et al. "Liver cell lines for the study of hepatocyte functions and immunological response." Liver International 25.2 (2005): 389-402" to test our compounds MOC-11, 23 and 24.

However, in the current unfortunate circumstances of COVID-19 pandemic, we have a limited ability to run laboratory work. I hope the respected Reviewer and the Editor may accept the manuscript as conceptual work without this experiment.

Reviewer 2 Report

This article focus on targeting the NS3/4A protease, which is a foremost-targeted protein in hepatitis C virus (HCV). The author in this article show compounds which can compete with NS4A on NS3 and resulting to NS3 inhibition.  Starting to de novo design and synthesis the NS4A peptidomimetics, then they evaluate series compounds by DSL binding assay, fluorescence anisotropy assay and enzyme inhibition assay. It is a well-prepared manuscript, provides enough background, detailed in the design and well presented. The study is well structured and easy to follow. However, I have following comments need to be cleared before acceptance.

For figure 3, it is suggest to convert positive control to 100 percent and show all others as percentage, this will give a more clear idea. The figure legend should not include too detailed methods.

Page 7 line 190, the concentration unit should be µM.

For the Enzyme Inhibition Assay, it is suggested to use the positive control from anaspec (highly active recombinant HCV NS3/4A protease), or dilute protein concentration to optimize the activity of your NS3. You should see a robust increase, but not only 9 fold. The line 211-212 did not make sense, it is not necessary to show activity in the absence of cofactors.

Author Response

Response to Reviewer 2.

Comment: This article focus on targeting the NS3/4A protease, which is a foremost-targeted protein in hepatitis C virus (HCV). The author in this article show compounds which can compete with NS4A on NS3 and resulting to NS3 inhibition.  Starting to de novo design and synthesis the NS4A peptidomimetics, then they evaluate series compounds by DSL binding assay, fluorescence anisotropy assay and enzyme inhibition assay. It is a well-prepared manuscript, provides enough background, detailed in the design and well presented. The study is well structured and easy to follow. However, I have following comments need to be cleared before acceptance.

Answer: We are pleased and feel grateful for encouraging this serious research work.

Comment: For figure 3, it is suggest to convert positive control to 100 percent and show all others as percentage, this will give a more clear idea.

Answer: Corrected as suggested and the related text has been also amended.

Comment: The figure legend should not include too detailed methods.

Answer: Legends was shortened and experimental details were eliminated not only from Figure 3 legend but also from Figures 4 and 5.

Comment: Page 7 line 190, the concentration unit should be µM.

Answer: Corrected

Comment: For the Enzyme Inhibition Assay, it is suggested to use the positive control from anaspec (highly active recombinant HCV NS3/4A protease), or dilute protein concentration to optimize the activity of your NS3. You should see a robust increase, but not only 9 fold. The line 211-212 did not make sense, it is not necessary to show activity in the absence of cofactors.

Answer: We have already reproduced the assay according to Anaspec manual, but the Anaspec’s protocol is designed for assaying competitive inhibitors, therefore results were not indicative. We tried to use diluted concentration of the enzyme but it did not work. The results were not consistent because some of compounds did not show any binding activities decreased the FRET protein fluorescence (these experiments and results were not included in manuscript). The NS3 concentration described in the manuscript produced nice activity. This method gave detectable catalysis in the absence of the cofactor. This result was confirmed by the binding assay methods (DSLS and FA) which use higher NS3 concentrations, since the binding and competition with NS4A was the primary objective of this work. It is worthy to mention that we published an earlier paper that presented peptide competitors of NS4A using the same enzyme inhibition assay (Ref 29). We would like to underscore that we truly intended to run the experiments once more as requested by the respected reviewer but the unfortunate circumstances of COVID19 pandemic, we are experiencing great difficulty to obtain material and to prepare fresh NS3 protein (or import it). Therefore, we kindly ask the respected reviewer and the editor to accept the qualitative enzyme inhibitions assay as described in the manuscript. Thank you

Round 2

Reviewer 1 Report

I accept the authors' response based on the current circumstances. They should state this explanation in the manuscript for readers.

Author Response

Comment: accept the authors' response based on the current circumstances. They should state this explanation in the manuscript for readers.

Answer:

We added the following statement in the Acknowledgment section:

Authors are exceptionally grateful to MDPI, the Editor Ms. Sophia Lin and the respected Reviewers for being very cooperative to publish this manuscript in the current circumstances of the COVID 19 pandemic that hindered us to cover some reviewers’ comments.

Thanks very much.
